# 3D Printed Energy Return Elements for Upper Limb Sports Prosthetics

**Jung Wook Park ***, **Ben Greenspan, Taylor Tabb, Eric Gallo and Andreea Danielescu**

Future Technologies R&D Group, Accenture Labs, San Francisco, CA 94105, USA
* Correspondence: jungwook.park@accenture.com

**Abstract:** Prosthetics are an extension of the human body and must provide functionality similar to that of a non-disabled individual to be effective. Sports prosthetics such as the Flex-Foot Cheetah from Össur have demonstrated the value of creating devices that both provide mechanical support and introduce passive energy return to mimic forces otherwise produced at joints. These energy return mechanisms have not yet been demonstrated for upper limb prosthetics but could improve their effectiveness and provide a greater range of motion and control. Using multi-material 3D printing technology, we extend energy return components to upper limb prosthetics by developing novel force-sensing springs and applying them to a basketball prosthetic. The 3D-printed springs compensate for the forces otherwise generated by wrist and finger flexion while measuring the mechanical deflection. We discuss design guidelines, methods for integrated 3D printed energy return within prosthetics, and broader applications in assistive technologies.

**Keywords:** upper extremity sports prosthetics; energy return; force-sensing spring; basketball prosthetic

## 1. Introduction

There are roughly 2 million people living in the USA with limb loss [1], and lower limb amputations are 10 times more common than upper limb ones [2]. Lower limb sport prosthetics have benefited greatly by incorporating energy store and return elements, while upper limb prosthetics designs are yet to see significant gains using this approach. Upper extremities typically perform higher dexterity and fine control tasks and have significantly more degrees of freedom than lower extremities. Despite this, existing upper extremity prosthetics are typically limited in functionality, particularly in sports, and often do not support more complex throwing motions and other similar activities. Simple movements remain challenging for users leading to frustration. Ultimately, up to 38% of individuals with upper limb loss choose to go without a prosthetic rather than using the ones available [3,4].

For athletes with upper limb loss, a variety of activity-specific prosthetics exist, but most function as adapters between the socket or wrist and equipment the individual is using, such as a baseball bat or a bike handle [5]. Athletes who play sports that require upper limb movements have few options for sport specific prosthetics. Research shows that individuals who return to playing sports after an amputation have improved psychological health, motivating the design of effective prosthetics to promote this return [6].

A prominent example of a sport specific prosthetic for the lower extremities is the Flex-Foot Cheetah from Össur which was worn by Oscar Pistorius in the 2012 Olympics [7]. This design uses a carbon fiber spring which stores the energy from the runner during the drive phase. This energy is then released at toe off to generate propulsion. Having the prosthetic act as a spring for this activity is very effective because running can be characterized as a spring-mass model [8], and the energy return element enables users to jump and run similar to a non-disabled individual. To our knowledge, no existing

upper extremity prosthetics, however, currently implement a energy storage and release mechanism. Instead, they fall into one of three control categories, which were designed for grasping: body-powered, myoelectric, or hybrid. In this paper, we propose *new energy return mechanisms to enhance upper extremity prosthetics* based on 3D printed force-sensing springs and cantilevers, for advantageous biomechanics performing specific activities.

Commercial prosthetics that are strong and lightweight are typically made of carbon fiber or titanium, such as the Flex-Foot Cheetah. These materials drive up the cost, with some advanced prosthetics costing over $15,000. The high cost associated with these has led several startups and non-profit organizations to create 3D printed prosthetics. 3D printing also makes it easy to customize designs to individual user preferences. Limited material choices have thus far prevented integrating energy return in 3D printed prosthetics. However, recent advances in multi-material 3D printing have enabled the potential for complex designs that are still cost-effective and tailored to an individual user [9,10]. This work leverages multi-material 3D printing to create two energy return mechanisms that could be included into these custom designs.

Basketball is an excellent example of a sport requiring upper limb movement that benefits from a higher degree of dexterity and control. It is also the most common ball sport played and the seventh most common recreational sport in the US [11]. The force applied from wrist and finger flexion on the ball is extremely important to control the speed, direction and spin of the ball when shooting [12]. Energy return mechanisms offer an opportunity to compensate for the forces generated by the wrist and fingers during a shot.

Inspired by the success of the energy storage and release designs of the Flex-Foot Cheetah, we investigated and evaluated methods to introduce energy return elements into upper limb prosthetics, specifically focused on sports applications. The purpose was to identify and demonstrate elements that simultaneously provided energy return and integrated sensing for feedback and evaluation purposes. The investigation focused on 3D printing methods, to keep the components accessible and to create individual elements that could be incorporated into custom designs. Characteristics such as compactness, ease of fabrication and range of forces generated were used as initial selection criteria. The results yielded two form factors that present practical energy return elements and include embedded sensing. The multi-material 3D printed coil spring and cantilever spring design are presented and characterized here. To illustrate a potential use case, we integrated both energy return elements in a fully 3D printed basketball prosthetic designed for transradial (forearm) amputations, which are the most common (47%) [13]. The example design illustrates the potential for energy return elements to be integrated within upper limb prosthetics to compensate for forces generated in a basketball shot or other sports applications. Embedded sensing is shown to provide real-time feedback on the forces present on each element which can be used for evaluation or integrated feedback to the user. We also discuss additional potential applications and the need for full user studies to evaluate the effectiveness of energy return in upper limb prosthetics.

## 2. Background

### 2.1. Energy Return in Prosthetics

About 90% of amputee paralympic runners use the Flex-Foot Cheetah, or a similar design that leverages energy return elements, in competition [14], while non-disabled runners have calves and ankles that return and amplify energy supplied by the hips and knees, the Flex-Foot design achieves a similar effect by mimicking a heel. Specifically, the Cheetah uses carbon-fiber reinforced materials to convert weight into energy during each step, enabling wearers to run and jump similar to a non-disable runner. However, studies indicate that the Flex-Foot does not completely replicate the missing limb joints. Therefore, users must develop adjustments in their stride and style, including generating increased energy from some muscles in contrast to a non-disabled runner [15,16]. These devices also enable many others with lower limb loss to enjoy a more active life

As seen with the example above, research indicates substantial benefits when prosthetics include energy return mechanisms [17,18]. The Vari-Flex, another design from Össur, is sold as a prosthetic foot designed to use energy return similar to the Flex-Foot Cheetah and provide a more natural gait while walking [19]. The Pro-Flex foot developed by Childers and Takahashi, improves upon the design by adding linkages that better direct energy return for users, increasing range of motion [20]. Other research has indicated that the benefits of energy return in lower limb prosthetics increase with adaptation, increasing walking speed and other metrics in many individuals [21]. Researchers have also investigated methods to create prosthetics similar to the Vari-Flex using lower cost methods to increase accessibility [22], while there has been considerable work in energy return for regular use lower limb prosthetics and footwear, the same is not true for upper limb prosthetics. Therefore, we draw inspiration from the benefits of energy return in lower limb prosthetics and apply them to upper limb prosthetics in our work.

## 2.2. 3D Printed Prosthetics

Because of the high costs of prosthetics, there has been a recent push to make them cheaper through 3D printing, which is low-cost and enables individual customization [23]. One such effort is the open sourced e-NABLE prosthetic which can be printed on any hobby grade 3D printer [24]. This prosthetic is for everyday use and translates wrist flexion into finger flexion, while this prosthetic hand can be used by anyone, Schmidt highlights its effectiveness for the developing world [25]. Companies like Open Bionics have also combined 3D printing with microcontrollers to create low-cost active prosthetics. 3D printing has also been used to make custom fitting orthotics and prosthetic sockets [26]. However, all of these are everyday use prosthetics and lack the specific fine motor function needed for certain sports [25].

## 2.3. Existing Basketball Prosthetics

A few commercial options exist for those who want to play basketball, mainly sold by TRS Prosthetics [5]. Two of these options are the Super Sport and Free Flex hands, which are general use flexible hands made from polyurethane elastomers that have a cupped smooth surface to make contact with a ball for a variety of sports. Another option, the Mill's Rebound Pro, features a wider curved surface specifically for rebounding. The Hoopster is another prosthetic hand designed for shooting and was conceived by an amputee, Hector Picard [5]. This simple design uses a hoop slightly smaller than the diameter of a basketball for a secure hold but smooth release. The downside of the simple design is that the wrist and fingers cannot flex like a human hand requiring significant changes in movement to perform a shot. To add backspin and increase accuracy, engineers from UCLA created 'The Spock' in 2016 [27]. The Spock is a non-commercial low-cost prosthetic basketball hand that was inspired by the e-NABLE. In contrast to the Hoopster, this prosthetic is for individuals that still have control of their wrist movement, although an adaptation that generates less force can be made for those that do not have this capability.

To our knowledge, these are the only prosthetics for basketball that exist today, none of which provide the user with any feedback on how much force is being applied to the ball nor the necessary mechanisms for compensating for missing muscles and joints. As mentioned earlier, it is still more common for individuals suffering from upper extremity limb loss to choose to play with no prosthesis than one of these options due to the limitations of the existing options on the market [3]. To address this, we draw inspiration from these designs and leverage 3D printed springs and sensing to incorporate both feedback and compensate for wrist and finger flexion to increase accuracy and control in our basketball prosthetic example.

## 3. Materials and Methods

### 3.1. Materials: Sensing with Conductive Polymer Composites

Conductive polymer composites (CPC) are mixtures of polymer and conductive material, such as carbon, that can be used in fused deposition modeling (FDM) printing [28]. The conductivity of these composites is dominated by a matrix of individual conductive pathways within the material, and small changes from stress, strain or temperature can be detected as a change in electrical conductivity [28]. The impact of these external stressors to the internal matrix of conductive pathways differs based on the material composition. Leveraging this phenomenon, Leight et.al. used a material termed "carbomorph", a composite of carbon black and polycaprolactone, to create 3D printed flex sensors that can be integrated with gloves and other applications to detect bending [29]. Gronborg et.al. demonstrated 3D printed force sensors based on cantilever beams with responses linked to the sensor dimensions, particularly the length of the bending sections [30]. This concept was also used to create touch sensing in a soft 3D prosthetic hand, in which the conductive polymer composite sensors were fabricated and applied to the fingers tips with a separate process [31]. These results highlight the potential of embedding force sensing directly into energy return mechanisms in 3D printed prosthetics. While sensing force or position with 3D-printed sensors has been demonstrated through a range of methods [32,33], our design leverages the properties of conductive polymer composites and printing techniques possible using FDM. Materials such as Proto Pasta conductive PLA (cPLA), which is a composite of PLA and carbon black, are readily available as filament for FDM printing.

### 3.2. Methods: 3D-Printed Springs

The accessibility and fabrication flexibility of 3D printing makes it an effective method of creating springs [34]. One example is Feron et al.'s leaf spring that was created through controlling its infill density and stiffness [35]. Other work establishes a programmable spring origami model using 4D printing [36]. 3D printing also enables seamlessly integration of a spring structure into a target object without any assembly. For example, Bodaghi et al. incorporated non-assembled spiral spring joints while designing a 3D-printed robotic finger and confirmed that 3D-printed springs could allow the rapid exploration and prototyping of different designs [37]. Recent work by He et al. presented a way of integrating a 3D-printed coil spring into everyday objects, which could be produced in a single print [38]. There has also been considerable work on fabrication techniques to support printing of complex designs such as springs [39,40]. However, to the best of our knowledge, research on transforming functional 3D spring models into force-sensing elements has not yet been conducted.

Multi-material printing with FDM allows us to easily adjust printing parameters (e.g., infill, layer height, grid pattern) and concurrently print conductive and structural materials [38]. Along with designed dimensions, altering the infill percentage and material composition directly affects the mechanical structure, and thus energy storage, of a printed coil spring. The typical rough surfaces left by FDM printing also provide an approach to sense forces between conductive elements. As contact force is applied to a cPLA surface, the surface features flatten and the contact area increases, reducing electrical resistance. Both the intrinsic change in resistance and the surface roughness of printed cPLA were investigated as sensing mechanisms for energy return coil springs.

All components were created and modeled using Autodesk Fusion 360. Models were sliced using Ultimaker Cura and printed on an Ultimaker S5 3D printer using PLA and conductive PLA (Proto-Pasta) using a standard 0.4 mm nozzle diameter. All parts were printed with a 90% infill density, a 90 mm/s printing speed, and a grid infill pattern. We used two print heads—one for non-conductive PLA filament and another one for conductive PLA. Printed springs were measured on a custom setup using an off-the-shelf force transducer (Mxmoonfree HP-500N, Figure 1), a digital caliper and a multimeter (Bolyfa 117 multimeter). Measurements were taken in increments of 0.25 mm with force and resistance recorded at each step. The resulting data was plotted to extract the measured

spring constant, $k$. Preliminary spring designs were evaluated with only a few testing cycles, while final designs were validated for repeatability through ten consecutive compression cycles to determine repeatability, hysteresis and signal error. Reproducibility was also assessed by comparing characteristics of two separate springs with identical designs. Initial design parameters were based on modeling and iteratively refined using component measurements.

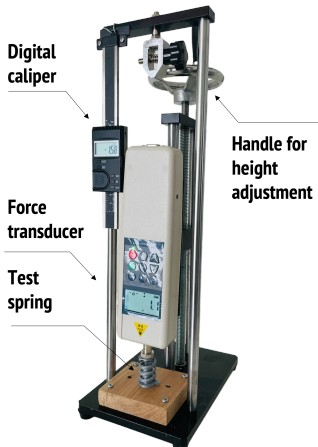

**Figure 1.** Experimental setup for the force-displacement measurement.

## 4. Energy Return Design

We investigated and iteratively designed two separate mechanisms for this work, a coil spring and a cantilever spring, to create functional energy return elements. The goal was to identify configurations and sensing mechanisms that would enable effective implementation of these energy return elements in an upper limb prosthetic. Designs were constrained to achieve compactness, be 3D printed, have adjustable parameters and provide the necessary energy return and force sensing signal for effective use. The developed designs should not be considered optimized but instead are intended to demonstrate how energy return components can be created and tuned for upper limb prosthetic applications using 3D printing.

Designing and fabricating a 3D printed spring can be a complex task due to varying material properties and printing constraints [38]. This is compounded when using more than one material or if additional features, such as deflection sensing, need to be integrated. Interfaces between different materials, variable mechanical properties of printed material and other factors can introduce differences between models and measured performance. This section serves as a guide to understand the principles of force-sensing springs used in energy return applications. We used modeling and an iterative design process to realize a set of components and discuss their design and resulting characteristics.

### 4.1. Coil Spring
4.1.1. Mechanical Structure

For mechanical analysis, Castigliano's theorem is a simple, yet powerful method to understand the deflection of a coil spring,

$$\delta_i = \frac{\partial U}{\partial F_i}, \tag{1}$$

where $\delta_i$ is the displacement from the point in which the force $F_i$ is applied, and $U$ is the total strain energy [41]. Specifically, the deflection-force relationship in a coil spring is represented by the equation,

$$\delta_{press} \doteq \frac{8F_{press}D^3N}{d^4G}, \tag{2}$$

where $\delta_{press}$ (unit: m) is the displacement when the force $F_{press}$ (N) is applied, $D$ is the mean coil diameter (m), $d$ is the diameter of the coil cross section area (m), $N$ is the number of active coils, and $G$ is the modulus of the coil's rigidity. (see Figure 2) The $G$ (MPa) modulus is equal to $E/2(1 + \nu)$ where $\nu$ is the ratio of the lateral and longitudinal strain, called Poisson's ratio, and $E$ (N/m$^2$) is Young's modulus. $G$ is material dependent and the rest of the variables are adjusted to achieve the desired force $F$ and meet space constraints.

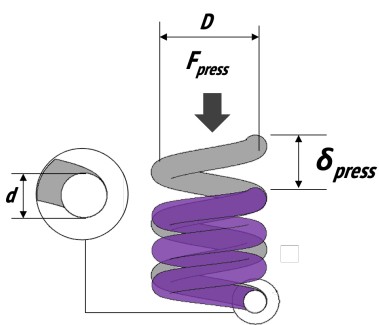

**Figure 2.** Mechanical structure of a coil spring.

4.1.2. Mechanical Energy Storage

The energy stored in a coil spring is called the elastic potential energy, $P_{elastic}$, and described by

$$P_{elastic} = \frac{1}{2}k\Delta x^2, \tag{3}$$

where $k$ is the coil spring constant, and $\Delta x$ is the deformation length of the spring. Through Equation (2), $k$ is defined as $k = \frac{F_{press}}{\delta_{press}}$. This stored elastic energy is later transformed into kinetic energy and used to return to its original shape. There is also possible energy loss during this transformation due to friction. Once material properties are known or estimated, coil springs can be designed to deliver targeted force or energy, and their volume and form factor can be minimized as needed based on the needs of the overall design. For instance, if designing an energy return element to simulate wrist flexion forces, a typical torque value of 5 Nm and hand length of 190 mm would roughly require a force of 25 N from the spring(s) in application [42].

4.1.3. Embedded Sensing

While sensing force or position with 3D-printed sensors has been demonstrated through a range of methods [32,33], our design leverages the properties of conductive polymer composites and printing techniques possible using fused deposition modeling (FDM). Materials such as Proto Pasta conductive PLA (cPLA), which is a composite of PLA and carbon black, are readily available as filament for FDM printing. The conductive material detects stress and strain through a change in material resistivity, as discussed previously. Multi-material printing with FDM allows us to easily adjust printing parameters (e.g., infill, layer height, grid pattern) and concurrently print conductive and structural materials [38]. Along with designed dimensions, altering the infill percentage and material composition directly affects the mechanical structure, and thus energy storage, of a printed coil spring. The typical rough surfaces left by FDM printing also provide an approach to sense forces between conductive elements. As contact force is applied to a cPLA surface, the surface features flatten and the contact area increases, reducing electrical resistance. Both the intrinsic change in resistance and the surface roughness of printed cPLA were investigated as sensing mechanisms for energy return coil springs.

4.1.4. Iterative Design of a Force-Sensing Coil Spring

We designed several coil spring configurations to investigate mechanical and electrical performance. To begin, we created a base spring design as shown in Figure 3a. This design included a prismatic joint and was made entirely of non-conductive PLA material. The design was used to validate fabrication methods and as a reference for subsequent spring iterations. We conducted theoretical evaluations coupled to models and empirical evaluations on the printed result to understand the effectiveness of modeling and verify that the spring would meet targeted performance. First, the dimensions presented in Figure 3a were substituted into Equation (2) to derive the theoretical values. In particular, the Shear modulus *G* was set to 1287 MPa referring to the material properties of PLA from a literature review [43]. The base spring was designed for a spring constant near 500 N/m. The fabricated spring had a measured value of 742 N/m and a maximum deflection of around 5 mm at 4.2 N applied force. We attribute the increased observed stiffness of the spring to printing resolution and material property estimates used in theoretical calculations.

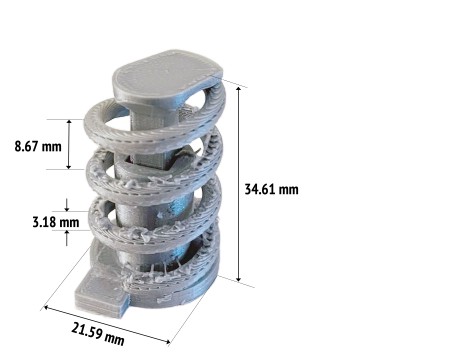

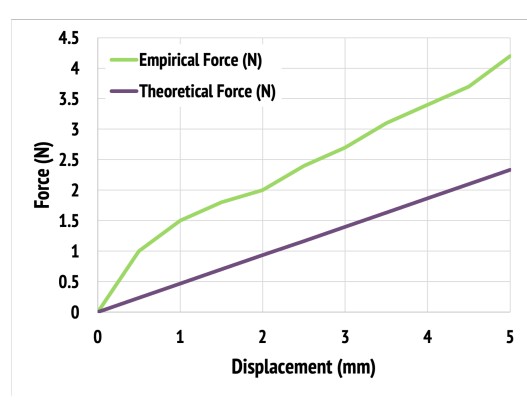

(**a**) Base Coil Spring          (**b**) Result of Base Coil Spring

**Figure 3.** (**a**) 3D-printed base coil spring and its dimensions, and (**b**) the results of force-displacement evaluation.

Using the base design as a starting point, we integrated cPLA and varied its shape, maintaining other spring design parameters as constant. The cPLA was introduced to enable embedded sensing, creating force-sensing springs. The first design iterations are shown in Figure 4. In Figure 4a, cPLA was inserted as the coil material. This design maintains a circular cross-sectional coil with a conductive spring and non-conductive linear prismatic joint. Force sensing is provided through the resistance change of the coil conductor. In Figure 4b, we implemented a partially conductive spring coil and a non-conductive prismatic joint, here the coils are modified to a rectangular sandwich structure. Force sensing is done by monitoring the conductive coil's resistance. Finally, Figure 4c, shows a design with a fully conductive coil and prismatic joint. Force sensing is measured through resistance changes in both the coil spring and the prismatic joint element.

Measurements were taken identical to the base spring, with the addition of recording spring resistance values at each displacement. A decrease in the spring constant was measured for each of the cPLA modified designs. Spring constants were measured as 230 N/m, 714 N/m and 480 N/m for Figure 4a–c, respectively. We attribute the decrease to the introduction of the cPLA and modified material properties. The variation of the material makeup and the geometry of the springs modifies the mechanical properties in relation to the base design. Material interfaces between the conductive and non-conductive materials and the different material properties of cPLA alter the G value of Castigliano's theorem. Maximum deflection remained similar for the modified spring or slightly smaller, between 4 and 5 mm for each.

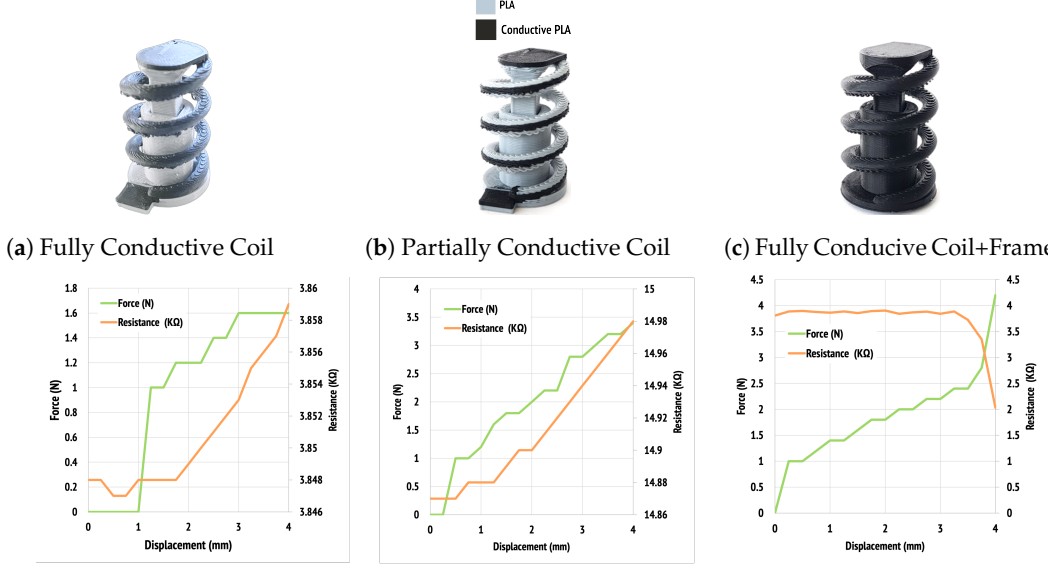

(**a**) Fully Conductive Coil  (**b**) Partially Conductive Coil  (**c**) Fully Conducive Coil+Frame

(**d**) Fully Conductive Coil  (**e**) Partially Conductive Coil  (**f**) Fully Conducive Coil+Joint

**Figure 4.** Variation of force-sensing coil spring design: (**a**) Conductive coil with non-conductive prismatic joint, (**b**) Partially conductive coil with non-conductive prismatic joint, and (**c**) Conductive coil with conductive prismatic joint. The force-displacement-resistance changes of each spring design: the result of (**a**–**c**) are (**d**–**f**), respectively.

The resistance change seen in both Figure 4d,e is linear and roughly proportional to applied force, with a observed change of about 1% of the total resistance at maximum deflection. Figure 4d shows an unchanging resistance below a 1.75 mm threshold, while Figure 4e provides a response across the full displacement distant. The dominant feature in Figure 4f is an abrupt decrease in resistance occurring when the trapezoidal geometry at the top of the spring makes contact with the top sleeve of the prismatic joint, at just over 3 mm displacement. This design provides closer to a threshold or binary output signal, rapidly decreasing past a certain level of applied force. Each design demonstrates the potential to include conductive elements within the spring for force measurement.

To further explore the surface contact sensing mechanism seen in the prismatic joint of Figure 4c, we designed and printed a constant pitch spring with a rectilinear cross-section but inverted the sandwich of Figure 4b and increased the coil height, creating two conductive surfaces that spiral down the coil as shown in Figure 5a. Resistance is measured between the top and bottom conductive paths, which operate as an open circuit when no force is applied. As the spring is compressed, the two conductors come into contact and a resistance change can be observed. This configuration resulted in a sharp change in resistance near its maximum compression, as all coils made contact simultaneously, bridging the top and bottom conductive paths along the full length of the coil. To create a more continuous signal change, we modified this design to a progressive pitch, Figure 5b. With a progressive pitch, the coils make physical contact ordinally from bottom to top, providing a continuously changing resistance value based on surface contact. The force-displacement of the spring was linear, providing a calculated value for the spring constant of 3376 N/m and a max displacement of 4 mm. The peak applied force was 13.8 N. This design produced a significantly higher spring constant than those of the base design.

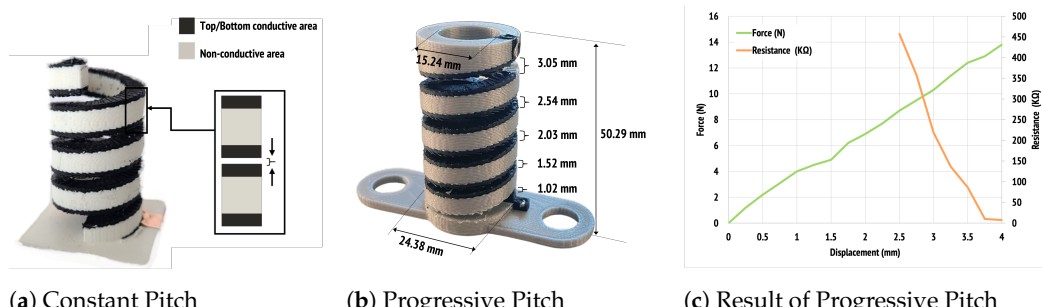

(**a**) Constant Pitch      (**b**) Progressive Pitch      (**c**) Result of Progressive Pitch

**Figure 5.** The different design of conductive layer to maximize the change of resistance—(**a**) constant pitch, (**b**) progressive pitch, and (**c**) the result of the progressive pitch model.

Resistance changes with respect to force were again observed in the progressive coil Figure 5c. An open circuit condition is observed below a displacement of 2.25 mm. Once contact is made in the lower coil, a sharp decrease in resistance in seen and a linear decrease of measured resistance continues as additional force is applied and displacement increases. At the onset of contact between the coils, the spring starts at a resistance of 457 kΩ when lightly compressed. When fully compressed, it has a minimum resistance of 7.3 kΩ. This spring has two observable output states as force is applied. Initially when the spring compresses and creates a connection between the upper and lower electrodes, there is a rapid decrease in measured resistance. As the distance between all coils compresses to zero, the rough deformable surfaces of the spring continue to compress, further reducing the resistance but at a significantly smaller slope. The spring also introduces substantial hysteresis that can be seen during rapid press and release of the spring.

4.1.5. Design Implications and Guideline

Each spring iteration provides a potential energy return element for upper limb prosthetics and other applications. Figure 4c, which incorporates a conductive prismatic joint can be designed to provide threshold force sensing, providing an alert if excess force is applied. By modifying the prismatic structure, coupled to the force displacement of the attached spring, the output signal can be tuned to trigger at a range of input forces. The design had a substantially lower spring constant compared to the base for energy return, with a calculated force potential of 1.9 N. The progressive surface contact spring, Figure 5, also provides resistance changes that can be mapped to applied force, with the progressive spring demonstrating a higher resolution signal and additional information on the spring state, such as 'in compression' vs. 'fully compressed' based on the force-resistance relationship. The changing resistance signal includes significant hysteresis under rapid repeated force cycles. The progressive spring also had the highest spring constant of 3376 N/m and a max displacement of 4 mm, with a potential peak force of 13.5 N which approaches the range of wrist forces. Both the fully conductive and the sandwiched conductive springs, Figure 4b,d provide proportional resistance change with applied force. Figure 4d provides a simple design with fewer issues with interfaces between conductive and non-conductive regions but the lowest measured spring constant of all configurations tested. Figure 4b provides flexibility for energy storage as the non-conductive components of the spring coil can be tuned to provide the necessary mechanical properties and parameters such as coil spacing can be chosen to match displacement requirements in a design. The mechanical properties were also closest to the base spring, potentially simplifying designs by relying on published values of non-conductive springs. Theoretical versus empirical values of key performance metrics for the base spring, such as the spring constant, did not match closely but theoretical values provided a reasonable starting point for a base design and iteration from the base enabled more accurate prediction of performance. We expect that improved material property constants for the as-printed materials would bring theoretical close to measured values.

### *4.2. Cantilever Spring*

#### 4.2.1. Mechanical Structure

Similar to the coil spring in Section 4.1.1, the deflection of a cantilever spring can also be understood using Castigliano's Theorem. The maximum deflection at the end of a cantilever is

$$\delta_{end} = \frac{F_{end}l^3}{3EI},\qquad(4)$$

where $l$ the length of cantilever (unit: m), $E$ is Young's modulus (N/m$^2$), and $I$ is the second area moment of the cantilever cross section. Note that $I$ is equal to $(bh^3)/(12)$, where $b$ and $h$ are the width and height of cantilever spring in meters, respectively, [41]. (see Figure 6a).

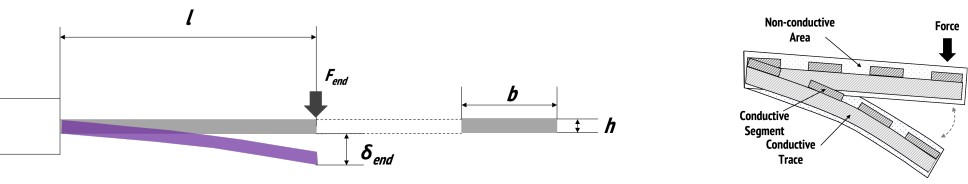

(**a**) Mechanical structure of a cantilever spring.        (**b**) Flex sensor design

**Figure 6.** (**a**) Mechanical structure of cantilever spring. (**b**) The structural change of flex sensor-embedded cantilever spring when force is applied at the tip.

#### 4.2.2. Mechanical Energy Storage

The strain energy, $U$, absorbed in the cantilever spring is defined as

$$U = \frac{F_{end}^2 l^3}{6EI},\qquad(5)$$

where the definitions of variables are identical to Equation (4) [41]. Like a coil spring, this stored strain energy is transformed into kinetic energy upon release and used to return to the original shape. However, Kathryn and Amos found that only 50.4% of the stored energy can be returned by the cantilever spring compared to an ideal energy storage-return model [44].

We envision cantilever-based energy return elements as mimicking lower force joints in a prosthetic, such as fingers, due to their simplicity and versatility. They can introduce forces for direction and guiding, such as when throwing a ball. Cantilevers can also be embedded within existing structures, providing a route to introduce energy return with minimal design changes and limited impact on overall size.

#### 4.2.3. Electrical Structure

The change in resistance due to bending in cPLA can also be integrated in a cantilever spring (see Figure 6b) This phenomena allow us to transform a mundane cantilever spring into a force-sensing mechanism capable of force sensing.

#### 4.2.4. Design of a Force-Sensing Cantilever Spring

Unlike the coil spring, we designed and examined our force-sensing cantilever spring in a single design. This is primarily due to the reduced complexity and design parameters of a cantilever. For this reason, our investigation concentrated on the sensing elements [45]. Mechanical properties can be controlled in a straightforward manner by adjusting the cantilever dimensions, while considering the documented energy loss typical in these structures [44]. We designed and fabricated the base structure shown in Figure 7a. The base cantilever had a 5 mm deflection at 4.8 N of applied force. Figure 7b displays the relationship between force and displacement, aligning well with Castigliano's Theorem as presented in Section 4.2.1. After validating the mechanical characteristics, we integrated the conductive elements with the base cantilever spring (see Figure 8a). The modified

cantilever demonstrated reduced stiffness with 4.1 N of force creating 5 mm of deflection and therefore reduced energy return. Fundamentally, the resistance value of a flex sensor varies according to its bending angle, and the cantilever spring's bend is determined by the applied force. By combining these unique characteristics, we were able to demonstrate a force-sensitive cantilever spring, with a force-resistance plot shown in Figure 8b. In this configuration, the resistance decreases as deflection increases.

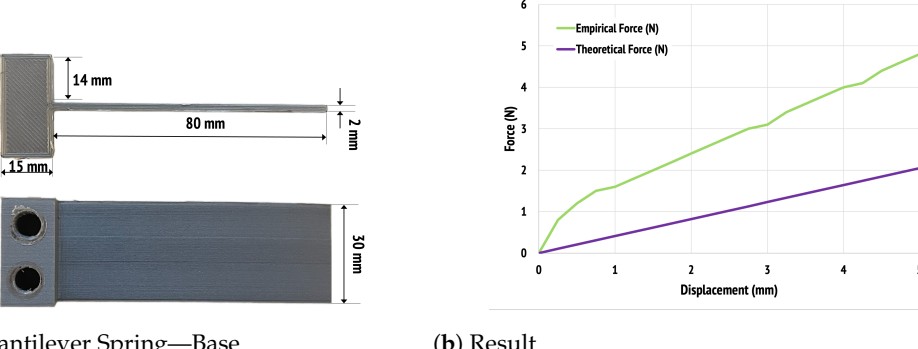

(**a**) Cantilever Spring—Base          (**b**) Result

**Figure 7.** (**a**) 3D-printed base cantilever spring and its dimensions. (**b**) the results of force-displacement evaluation.

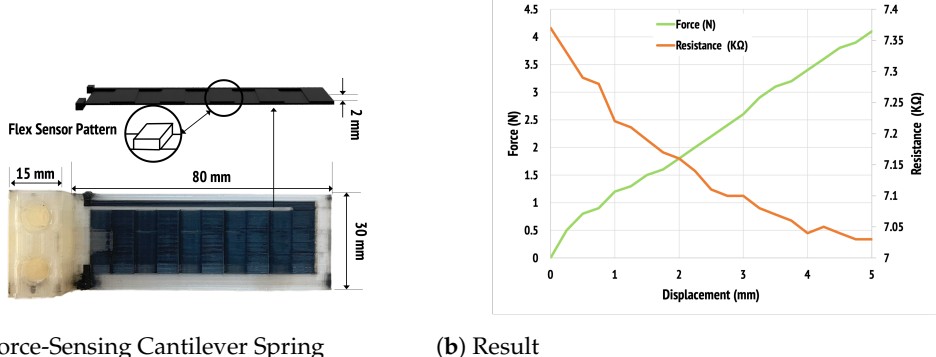

(**a**) Force-Sensing Cantilever Spring          (**b**) Result

**Figure 8.** (**a**) 3D-printed force-sensing cantilever spring and the shape of embedded flex sensor, and (**b**) the results of force-displacement-resistance evaluation.

### 4.2.5. Design Implication and Guideline

Design of the cantilever energy return mechanism produced expected outcomes for mechanical and sensing properties. The simple design of a cantilever allows adjustments to its spring constant and energy storage capacity through changes to its dimensions once a base design is empirically measured or material properties are well known. Introducing cPLA for sensing was shown to decrease this energy storage capability by a small amount but no optimized configurations or geometries were explored for the cantilevers beyond validation of energy storage and sensing capabilities. Introducing cPLA enables force and displacement detection through resistance changes in the conductive strip. A cantilever's energy storage and force sensing can be widely tuned to match mechanical and electrical requirements.

## 5. Bio-Mimetic Prosthetic Design with Force-Sensing Springs

As a demonstration, we integrated our energy return components, the force-sensing coil and cantilever springs, into a customizable prosthetic basketball hand for shooting. For the design, we researched the biomechanic principles of basketball shoots and then incorporated energy return elements to compensate for wrist flexion and finger motion in the prosthetic hand. We chose target performance values for energy storage and return force based on available literature to achieve a reasonable initial design. Since each user's

mechanics and generated forces might be different, we developed and evaluated a tool that allows them to capture and analyze their shooting motions through energy storage and return data during use. The design illustrates a potential application of the energy return elements in a fully 3D printed prosthetic.

### 5.1. Fundamental Biomechanics

In this instance, the coil springs were targeted as compensation for wrist motion during a shot while the cantilever springs were positioned to compensate for fingers as the ball leaves the hand. There are many variables that contribute to a successful basketball shot. The most critical from a physical perspective are the release height, release velocity and angle of release [46,47]. Lower limbs are primarily responsible for shooting distance, while upper limbs provide fine tuning of shooting movements [48]. Shot accuracy is determined by a kinematic chain that includes multiple degrees of freedom: shoulder, elbow, wrist and finger flexion, which aim and release the ball to create the proper angle and introduce rotation in the shot [12,47].

Immediately before the forward motion of the shot begins, the shooting arm is in a loaded position. This position has the shoulder above $90°$, the elbow in flexion, the wrist in almost full extension, and the fingers in close to full extension but in contact with the ball. To generate the forward and upward velocity required to aim and shoot the basketball, elbow extension occurs first, increasing the forces applied by the hand on the ball, with wrist flexion shortly after, providing vertical and horizontal forces, respectively. As the elbow continues to extend and the wrist flexes, the ball rolls closer to the fingertips causing a small amount of finger extension. Finger flexion applies the last bit of force to the ball as it leaves the shooter's hand, generating backspin for accuracy. The index and middle or middle and ring fingertips are the last two points to make contact with the basketball. By applying different amounts of force with each finger, the player can make very small corrections that affect the trajectory of the ball [49].

Existing literature reports that for a medium length shot (4 m, roughly the distance of a free throw), the basketball is typically released with a speed of 6.62 m/s at an angle of 46.0 degrees and $4\pi$ rad/s of spin and all shots follow an arched travel path to the basket, with varying release angles and velocities depending on shot conditions [50]. Literature indicates that this coordination varies based on shot distance, age, skill level and context [47,48,51]. Average shooting requires total forces between 17.5 N to 38.8 N [46]. These force values include those generated by all joints in a shot. Additional research indicates that a human wrist can generate torque roughly between 5 and 15 Nm during wrist flexion [42]. We chose the lower range of 5 Nm to base our designs. In players using wheelchairs during play, Malone et al. indicated that kinematics are adjusted according to physical limitations, with players capable of lower release heights using increased release velocities and steeper ball trajectories [52]. Most studies indicate that while the fundamentals of joint movement and force generation remain the same, players adapt their movements according to their abilities and conditions. This adaption ability is similar to those seen in lower limb sports prosthetic users, such as with the Flex-Foot Cheetah [15,16].

### 5.2. Prosthetic Energy Return Design

To attempt to mimic the biomechanics above, we designed a customizable basketball hand which attaches to a standard prosthetic wrist. As mentioned, the force applied to the ball by the hand from wrist flexion creates motion towards the basket; thus, the force is roughly parallel to the horizontal plane when released while fingers introduce spin and provide the last point of contact prior to release [12]. Both wrist and finger flexion are active movements where muscles contract to generate the necessary motion and apply force to the basketball. We aimed to create a design that would allow an individual with transradial limb loss to more naturally shoot a basketball. To accomplish this, we integrated two energy return coil springs attached to a hinged component that is curved to hold a basketball that compensates for wrist flexion and two cantilever springs to mimic fingers.

The cantilever springs are offset from the main load path to delay the force applied by finger flexion, replicating a natural shot. The coil springs are compressed as the shooter moves upward, generating force primarily from their legs. As the user pushes the ball upward, and the coil springs release their stored energy, the ball rolls up to the cantilever springs forcing the cantilever spring to bend backwards. When the user's arm begins to slow, the cantilever spring applies force to the ball creating backspin. Because the index and middle, or middle and ring, fingers are the last two points to make contact with the basketball, our prosthetic has two fingers. Using bilateral springs and cantilevers also reduces the require force production from each individual component.

Focusing on the hand and wrist, a basketball shot consists of three phases: load, roll to fingertips, and release. In the load phase of the shot, as the elbow begins to extend, the wrist is in near full extension (Figure 9(a1)). In our design, the coil springs are compressed, causing the ball to sink deeper into the prosthetic just like it would sink deeper into the palm of the shooter's hand (Figure 9(b1)). As the wrist flexes, the ball rolls up to the shooter's fingers as a small amount of finer extension occurs (Figure 9(a2)). In our design, as the shooter's arm begins to decelerate, the coil springs release causing the ball to roll up to the cantilever fingers, which extend (Figure 9(b2)) Finally, two fingertips provide the last points of contact with the ball. The finger flexion exerted by these figures on the ball are what generates backspin (Figure 9(a3)). The prosthetic functions the same way with the cantilever springs releasing, redirecting the force on the ball to generate backspin (Figure 9(b3)). The cantilever springs were designed to match the size and placement of a shooter's hand on the ball (Figure 10).

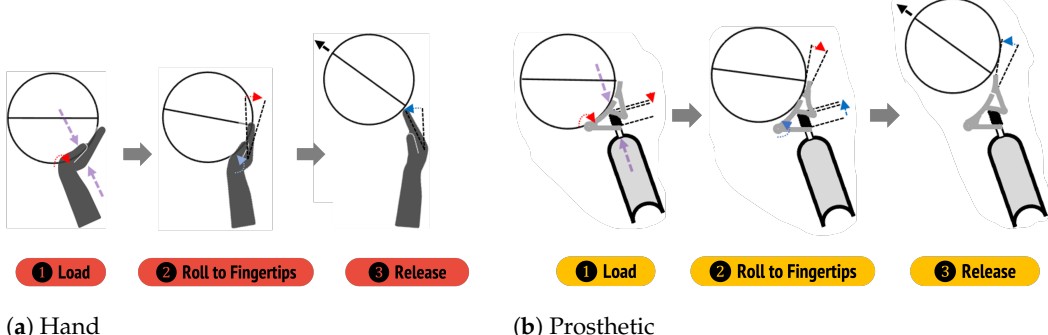

(**a**) Hand                                                                                     (**b**) Prosthetic

**Figure 9.** Hand movements in a typical overhead basketball shot (**a**) and their analogs in the presented prosthetic (**b**).

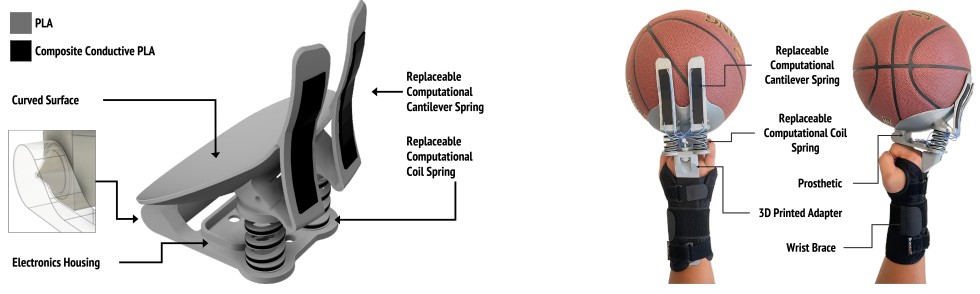

(**a**) Final Design of Prosthetic                                            (**b**) Prosthetic Attachment

**Figure 10.** Individual components of the final prosthetic (**a**); the structure of our final design (**b**); final prosthetic attached to wrist brace via 3D printed adapter.

For the prosthetic, two modified spring designs were implemented to both match estimated forces and to enable integration with the 3D printed prosthetic (see Figure 11). The coil spring design was based on the sandwich configuration, Figure 4b. This configuration was chosen for its linear sensing signal over the full range of displacement and its larger maximum deflection. The updated design targeted a maximum applied force of 12.5 N for

each of the two coil springs implemented in the prosthetic. The fabricated springs were measured to have a spring constant near 900 N/m and a maximum displacement of 14 mm, providing a calculated force return (not considering losses) close to the 12.5 N target value. Modifications to the spring included adjusting its dimensions to fit within the prosthetic frame and modifying the coil dimensions to provide the increased spring constant and maximum displacement as shown in Figure 12a. The cantilever design was also modified to estimate the dimensions of a finger. Force-displacement measurements showed a maximum displacement of 40 mm at a force of 10.7 N as shown in Figure 12b. Measurements were repeated on a single spring over ten compression cycles to characterize repeatability and identify potential hysteresis within the structure. Both mechanical and electrical results became consistent after two compression cycles, providing repeatable force (standard deviation < 0.4 N) and resistance (standard deviation < 30 Ω) results versus compression distance and no observable hysteresis. Extended compression cycle performance beyond these ten sequential measurements was not assessed due to instrumentation limitations. Measurements between two printed springs with identical designs showed relatively repeatable mechanical characteristics, with a difference of roughly 1 N at peak displacement. The springs however demonstrated more significant differences in the absolute resistance value of the coil springs and the slope of the resistance-displacement curves. For example, the resistance values of two coil springs printed in the same geometry and the same printer settings were 7.8 kΩ and 8.2 kΩ, respectively, when not pressed. An initial characterization and calibration are required for each spring to determine the force-resistance relationship for sensing. It is important to note that these designs should be regarded as preliminary estimates of the required parameters for energy return in the prosthetic and user testing would be required to determine actual optimal values, including measurements of energy losses within the springs and cantilevers.

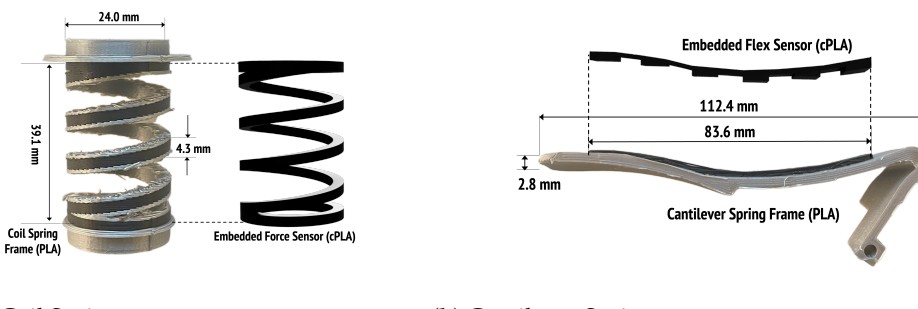

(**a**) Coil Spring    (**b**) Cantilever Spring

**Figure 11.** The force-sensing coil (**a**) and cantilever (**b**) springs used in our prosthetic design.

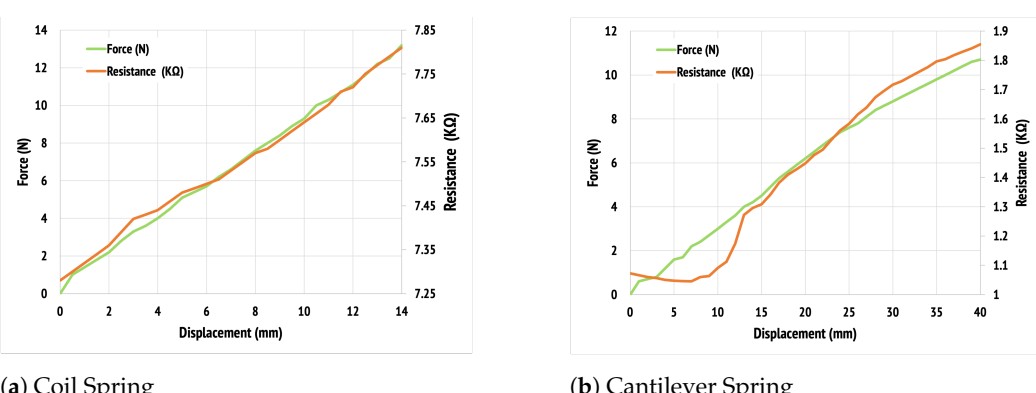

(**a**) Coil Spring    (**b**) Cantilever Spring

**Figure 12.** The results of force-displacement-resistance evaluation of coil and cantilever springs used in the proposed prosthetic.

The final prosthetic prototype attempts to mimic the natural motion of a basketball shot and can be easily printed in three parts: the main body, two force-sensing coil springs,

and two force-sensing cantilever springs (see Figure 10). It has an integrated printed conical joint that the prosthetic rotates about, just like the wrist. The main body contains an integrated space to house an electrical sensing system that receives signals from each energy return element. The entire prosthetic weighs a total of 141 g, making it lightweight for use. The design allows energy return elements to be exchanged or replaced, enabling adjustments to the energy storage and force generation values.

Physical limitations prevented exact replication of wrist flexion with passive energy return. Our design instead introduces force in line with a typical shot at 45° as opposed to the roughly forward force generated by a wrist during a shot. The coil springs are attached to hinged component that holds the ball during a shot. As energy is released, the springs rotate this component about its hinge, replicating wrist flexion. The hinged component rotates to a maximum angle of 45°, limiting the direction in which final force is applied. It is expected that this will require a modification to a players form, as seen with players with other prostheses [19,52]. The two coil springs together can produce a force of roughly 25 N (neglecting losses), comparable to estimates from the literature. The cantilever springs are expected to produce a force in the range of 2–4 N each dependent on the displacement during a shot. Energy storage and peak force values were chosen as preliminary estimates for demonstration and user validation will be required to optimize the designs.

### 5.3. Real-Time Monitoring System

To support customization and modification of a design including energy return, we developed a real-time monitoring system composed of a hardware sensing device and a software application. Both energy return spring designs can be integrated similarly to standard analog sensors. The only additional circuit element required for readout is a voltage divider, with the output of the divider fed into an analog-digital converter (ADC). The included microcontroller (nRF52832) has two cores—one for collecting ADC values and packetizing them and another for transmitting the packet to the connected mobile device over Bluetooth Low Energy. The electronics are compact and integrated directly into the prosthetic for mobile use. An Android-based application records the shooting motions through a rear-facing camera while visualizing and logging the force data transmitted from the hardware device. The application includes the necessary calibration data to translate the analog signals to measured force values. (see Figure 13) A user can record real time signals during use to better optimize the design of energy return elements within the device. This data, coupled with information collected about the ball path during shots can allow designers or users to tune the energy return parameters of the springs for the individual needs or applications.

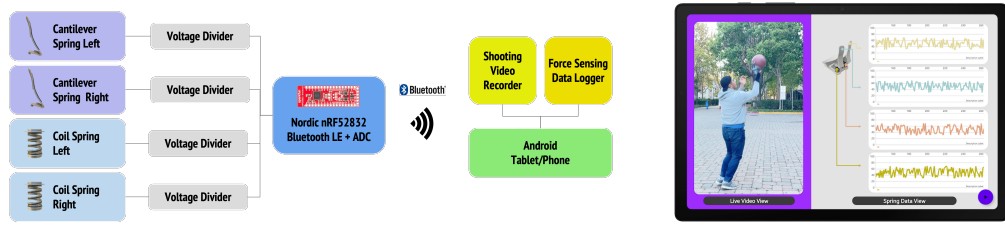

(**a**) Block diagram                    (**b**) Mobile application

**Figure 13.** Block diagram (**a**) and mobile application (**b**) of Shooter's Touch. This system allows the shooters to record changes in force in real time while recording their actual motion.

A basketball shooter can receive feedback on how much force is being applied to the ball through their palm and fingers. Information on ball orientation and applied forces is critical for making fine adjustments to the ball mid-shot, such as applying more force through one finger over another, to increase accuracy. These applied forces can be mapped to generate haptic or other feedback during use. Feedback can be custom configured using the four available signals to best inform a user during a shot. For instance, overall balance

of a shot can be reported by taking a constant differential of the bilateral coil springs. In a haptic feedback implementation, vibration can indicate forces by positioning motors on either side of the arm for example. As the force sensor signals can be easily mapped to outputs, it also allows individual athletes to build a model of how they would like feedback delivered for the most accurate shot.

*5.4. Evaluations*

Similar to the evaluation of other prosthetic research devices [53], we created an adapter to allow a non-disabled individual to shoot a basketball and test the performance of the prosthetic. The adapter consisted of two components, a wrist brace, like the TRS Prosthetic Simulator [8], and a 3D printed extender. The distal end of the extender attaches to the prosthetic via the same universal threaded stud of a standard wrist unit. The proximal end of the extender has a thin, flat form factor so it can slide through the Velcro straps of the wrist brace, securely attaching it to the forearm.

Our basketball prosthetic was created as a design example to illustrate how energy return elements might be integrated into upper limb prosthetics and was not fully evaluated in user studies. As a preliminary evaluation and validation of the energy return elements and force-sensing springs two separate authors tested the prosthetic using the adapter. The first tests were for shot performance, taking a large set of practice shots from the free throw line. Following practice, 6 out of 10 shots were made. Testers had difficulty adjusting the adapter for the prosthetic to fit comfortably but otherwise found the system easy to use and adjust to. It was observed that the shooter was able to adapt their form and achieve the expected trajectory of the ball towards the net. These adaptations were without feedback from the sensors. In a second test, the prosthetic was connected to a real-time monitoring system and force sensing values were collected during a shot. The visual sequences showed the applied forces on the spring coils and cantilevers as the ball leaves the prosthetic, Figure 14. We also confirmed it by analyzing the log data as shown in Figure 15. Note that the two y-axes were used in Figure 15 to display both sets of data on one graph because the relative values of the force data are quite different. The feedback could be used to improve and streamline the user's adaption during shooting, providing finer feedback of the prosthetic's dynamics, while these designs provide all the tools needed to implement feedback to a user, these benefits would need to be validated and tuned in full user studies of the device. All validation and testing involving the authors was done in compliance of the human testing protocols of the private institution at which the research was performed.

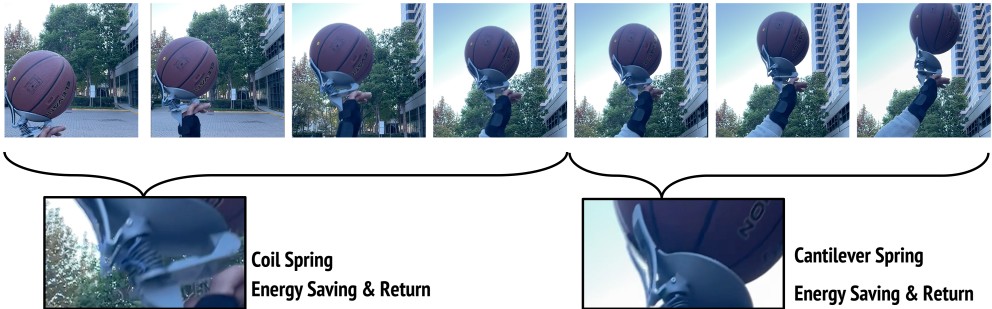

**Figure 14.** The sequence of shooting with two energy storage and return devices—coil and cantilever springs—in the proposed prosthetic.

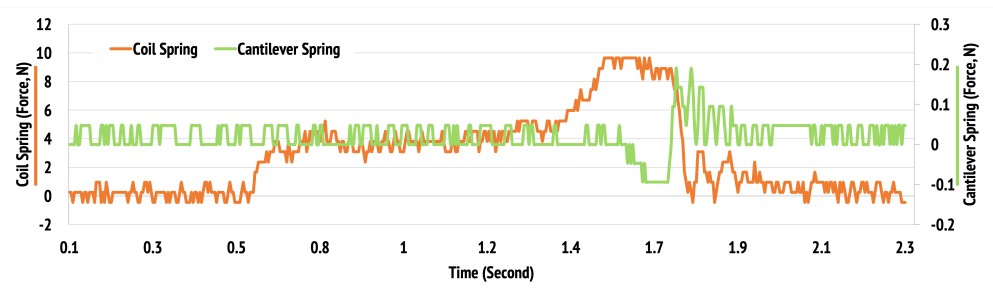

**Figure 15.** The results of force estimation from the data collected Shooter's Touch system. Left Y-axis is the force of coil spring. Right Y-axis is the force of cantilever spring.

## 6. Discussion

### 6.1. Energy Return in Prosthetics and Beyond

In this work, we investigated the feasibility of creating 3D printed energy return elements with embedded sensing for upper limb prosthetics. We identified, evaluated and demonstrated two compact 3D printed designs in the form of coil and cantilever springs that can be tuned for energy storage and form factor. Incorporating conductive polymer composite materials extended the functionality of the components to force-sensing springs, providing real-time output of applied forces through a resistance change within the coil or cantilever. The designs offer a 3D printed mechanism to both incorporate and sense applied force in a prosthetic with the goal of mimicking joints such as wrists and fingers, particularly in sports. Incorporating sensing elements was found to impact the mechanical properties to varying degrees in each configuration. 3D printed spring designs are able to deliver forces comparable to wrists and fingers, while maintaining compact sizes and small weights. These factors, combined with their inherent customizability, make the energy return components described here excellent candidates for implementation in upper limb prosthetics. 3D printing did not enable a high degree of repeatability between prints for sensing, but each spring demonstrated repeatable characteristics after an initial set of two compressions. The results and designs are intended to provide a qualitative overview of potential configurations for energy return springs providing embedded sensing. Furthermore, they were not optimized to maximum spring constant or sensing signal strength as each application will have its own set of constraints and requirements.

We demonstrated the potential for integration of these components in a basketball hand prototype, fabricated through multi-material 3D printing, while our force-sensing spring designs were targeted and applied to an upper limb prosthetic device, the design principles are applicable to general prosthetic applications and beyond. Sports such as baseball, bowling, and frisbee all require fine control of wrist and finger motions. In these applications, custom energy return designs could be implemented to generate the appropriate amount of force during play while providing force feedback signals for fine tuning or haptics integration. Outside of sports, there are other applications where an energy return system could used in upper limb prosthetics. Actions such as precision grasping often move very slowly. Adding an energy return component may improve performance and make various daily life activities easier to achieve. A few examples include being able to push open a heavy door more efficiently, quickly generate extra grip strength to turn an otherwise stuck lid or pressing a button at a challenging angle. By understanding the force dynamics in the energy return components, they can be designed to store and release energy when needed.

For non-prosthetic applications, augmenting an individual's performance with an energy return feature has already been shown to be effective in commercial products. A few footwear companies are beginning to 3D print the mid-soles of their shoes to increase the energy return during walking or running [54]. Designers have focused on optimizing material structure to achieve the greatest impact on absorption and energy return [55]. Using energy return shoes as an example, pressure distributions can be recorded from the runner's stride through the springs themselves without the need of an external sensor by

instrumenting force-sensing springs into the midsoles with a conductive polymer composite material. Using this information, the timing of the energy release can be optimized for greater energy return and may lead to more accurate step counts and activity classification. Additionally, real-time sensing could be provided to the user, which may help improve a user's stride and avoid injury.

Recent studies have yet to reach a conclusion as to which is superior between body-powered and myoelectric-based prosthetics. The former is well suited to applications that require a lot of energy, such as sports, and the latter is better optimized for fine adjustments, such as grasping an object [56]. In general cases, prosthetics are designed using one of the two methods, but a basketball prosthetic is a unique example requiring their combination. In our proposed prosthetics, the coil spring replicating the wrist motion is designed to provide more force, while the cantilever spring imitating the finger must be capable of finer adjustment of forces. More accurate basketball shots might be possible using a combination of energy return and other approaches, for instance if the force-sensing data from the cantilever can be incorporated into the myoelectric-based approach with an actuator.

### 6.2. Diverse Materials Available for Advanced 3D Printing

In our initial prototype we focused on using two different materials—PLA and conductive PLA. However, there are 3D printers that have more than two print heads, such as the Stacker S4, which can allow for the use of more materials in the same print. This could be leveraged to add a filament for additional flexibility to the fingertips that would further mimic how a human finger bends or additional elasticity in the springs. For example, thermoplastic polyurethane (TPU), which is much more flexible with a flexural modulus of 79 MPa, compared to PLA's high modulus of 3150 MPa, could add additional functionality to a prosthetic device [57,58]. Generally, flexible filaments also have more friction than plastics, providing additional grip where used. Polyvinyl alcohol (PVA) or a dissolvable support material can also be used to print more complex geometries. A force-sensing spring able to withstand higher forces for applications such as lower limb prostheses using polycarbonate may also be possible [59].

### 6.3. 3D Printed Components

The energy return elements described and evaluated here can provide accessible and inexpensive primitives that the community of 3D printed prosthetic designers can use to further enhance and customize upper limb prosthetics. The materials widely used in 3D printing provide an alternative to the carbon fiber and other expensive materials in commercial prosthetics, while embedded force-sensing elements eliminate the need to create methods of integrating external sensors. After material properties are empirically established, designers can use models to adjust the springs to meet force, size and sensing needs rapidly. The force sensing components can also easily connect to inexpensive and compact micro-controller boards to provide a customizable interface to an energy return system. The embedded force sensing also provides a route to rapid optimization of the design through user testing and evaluation. Overall, both energy return elements provide flexible approaches that can be designed and optimized quickly for customized prosthetics.

### 6.4. Limitations

Although we focused the introduction of the concept of energy return components and their application to prosthetics in this paper, rigorous validation with users are absolutely necessary for prosthetic design. Our energy return elements are designed based on available data in the literature but specific information regarding values such as 'wrist force and torque produced in a basketball shot' are not always specifically available. Therefore the results presented here are based on estimates of necessary performance. In the future, we plan to examine our prototype through a series of user studies and testing to gain insights on the role of energy return in the prosthetic's performance and tune the design to individual needs. This exercise will be coupled with data collection and optimization of the

devices to best match user preferences. With limited user testing, no long-term durability data could be gathered to understand how these devices will perform over a longer time. An extended evaluation of the components that couples to enabling users to adapt to the prosthetic and adjust motion mechanics would also strengthen the design and provide additional insight on modifications needed to create an effective, durable prosthetic.

*6.5. Future Work*

The energy return components focused on the design of two well characterized and simple spring elements, a cantilever and coil spring. Using the properties of conductive polymer composites and multi-material 3D printing, we believe there are many additional configurations that can provide simultaneous energy return and sensing for upper extremity prosthetics and other applications. These configurations may be more optimal for mimicking joints and seamlessly integrating with lightweight compact prosthetics. Further characterization and refinement of the energy return designs will also be completed along with an assessment on their potential in additional prosthetic energy return applications. The design will be evaluated for long term repeatability in both mechanical and electrical function. Exploring spring design tools to tune both the mechanical and electrical properties will enable higher levels of customization for individual users in basketball and other sports. The concepts demonstrated may also have applications in other non-spring mechanisms where force measurement is of value.

Our example prototype focused on optimizing a basketball shot. The application was chosen to best illustrate both energy return elements. Further design modifications and investigation are required for other aspects of playing basketball, such as dribbling, where energy return elements may also improve performance. Future work will need to test the current design's effectiveness for these other activities and further optimize the design so that it can support all of the activities a user may perform during a basketball game. Finally, we will look to apply energy return technology to other prosthetic sports and non-prosthetic applications.

## 7. Conclusions

We investigated and designed 3D printed energy return components with embedded sensing and demonstrated their potential use in a customizable basketball prosthetic hand. Along with compact energy return, we developed a method to transform 3D printed conventional springs into force-sensing elements and examined their characteristics. We fabricated an example design and its spring components using a multi-material 3D printing technique, combining PLA and composite conductive PLA. This is the first upper extremity prosthetic design that we know of that includes an energy return system used this way. Energy return elements enabled a prosthetic design that more closely emulates some of the motions of a human hand for a more natural basketball shot, and its force-dynamic data was presented to the user through a mobile application. The force sensing spring coils and cantilever designs can be customized and used in additional 3D printed upper limb prosthetics or other designs to increase performance or add functional elements. The concepts presented will pave the way for more effective and natural upper limb prosthetics to those that need them.

**Author Contributions:** Conceptualization, J.W.P., B.G. and T.T.; methodology, J.W.P., B.G., T.T., E.G. and A.D.; software, J.W.P.; initial validation, B.G., T.T., E.G. and A.D.; formal analysis, J.W.P. and E.G.; writing—original draft preparation, J.W.P., B.G., E.G. and A.D.; writing—review and editing, J.W.P., E.G. and A.D.; visualization, J.W.P.; supervision, E.G. and A.D.; project administration, J.W.P., E.G. and A.D. All authors have read and agreed to the published version of the manuscript.

**Funding:** This research was funded by Accenture, LLP.

**Institutional Review Board Statement:** The study (*Accenture Basketball Prosthetic*) was conducted in accordance with the legal and ethical requirements of Accenture, LLP.

**Informed Consent Statement:** Informed consent was obtained from all subjects involved in the study.

**Data Availability Statement:** Not applicable.

**Conflicts of Interest:** Jung Wook Park, Taylor Tab, Eric Gallo, and Andreea Danielescu are currently and were employees of Accenture at the time research was done. Ben Greenspan was an employee of Accenture at the time work was completed but is now an employee of another company.

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
