# Peer review of "3D Printed Energy Return Elements for Upper Limb Sports Prosthetics"

_prosthesis, doi:10.3390/prosthesis5010002_

Round 1

Reviewer 1 Report

The paper presents design and trials to include energy return elements into upper limb prostheses. Inspired by the positive experience of including such mechanisms into lower limb prostheses, the authors investigate how to provide a similar improvement into arm prostheses, which even when actuated only use the force exerted by the motors.

The challenge is investigating whether adding force sensing springs to compensate for forces generated during the movement by wrist and fingers flexion can improve the performance and satisfaction of the user.

Two solutions are proposed, based on coil springs or cantilever springs. The task of manufacturing them and inserting them into the prosthesis is made affordable by using 3D printing, exploiting conductive polymer composites. After studying the mechanical model of energy storage and the possible use as force sensing, the paper presents prototype implementations and experiments.

The specific task considered is basketball. The final case study evaluates the two kinds of springs according to a biomimetic analysis of the forces exerted by human arms when shooting the ball. The experiments are made on a prosthesis for trans-radial amputation with 2 fingers, integrated with 2 coil springs to compensate for wrist flexion and two cantilever springs to mimic fingers. Qualitative evaluations obtained from healthy people wearing the prosthesis conclude the paper.

As the authors clearly indicate, the paper is a first step in the task of designing more efficient prostheses. As such, it presents some trials, more or less effective, and a preliminary design. Anyhow, the results here presented are of interest and can inspire future developments.

The paper is well organized and the practical aspects of the various design phases are enough described. Even though the application task is very limited, the results are of interest. Applying it to every day life tasks is still a long road, as reported by the authors in the conclusion section, which also presents a reasonable road to continue the research.

Before publishing, a little extension is suggested. As the authors are interested in increasing user satisfaction, they could add some considerations about it on the basis of the preliminary tests. The monitoring system could be better described. The opinion of the users of the device could be more reported.

Author Response

We thank you for your constructive comments, which have led to an improved manuscript. To discuss the aspect of user satisfaction, we added the testers’ feedback to Section 5.4. Additionally, we described the technical details of the monitoring system in Section 5.3

5.4. Evaluations

[Line 581] Testers had difficulty adjusting the adapter for the prosthetic to fit comfortably but otherwise found the system easy to use and adjust to. It was observed that the shooter was able to adapt their form and achieve the expected trajectory of the ball towards the net. These adaptations were without feedback from the sensors.

5.3. Real-time Monitoring System

[Line 541] To support customization and modification of a design including energy return, we developed a real-time monitoring system composed of a hardware sensing device and a software application. Both energy return spring designs can be integrated similarly to standard analog sensors. The only additional circuit element required for readout is a voltage divider, with the output of the divider fed into an analog-digital converter (ADC). The included microcontroller (nRF52832) has two cores—one for collecting ADC values and packetizing them and another for transmitting the packet to the connected mobile device over Bluetooth Low Energy. The electronics are compact and integrated directly into the prosthetic for mobile use. An Android-based application records the shooting motions through a rear-facing camera while visualizing and logging the force data transmitted from the hardware device. The application includes the necessary calibration data to translate the analog signals to measured force values. (See Figure 14)

Reviewer 2 Report

The paper describes an application of force-sensing springs, which are applied to a basketball prosthetic.

It would be good to pose the contributions as scientific hypotheses / research questions. State clearly what the questions are that will be addressed. The contributions that are not scientific questions, should be described in the discussion.

The measurement errors should be mentioned, as at the moment the results are presented without these being taken into account.

It would be good to include calibration curves and number of repetitions taken to establish each.

In most cases, it is unclear how repeatable the results are, as the shown results do not include any repeat measures.

It would be good to comment on any e.g. drift and hysteresis that is present.

How consistent are the prints? Is there variability between similar prints? The images show that the print resolution leads to added/missing material in certain areas. What is the effect of this? In general, more information is needed to describe the accuracy of the prints.

Can it be assumed that the the user can adapt, due to changes in motor control based on the real-world outcome instead of the feedback system (as for example suggest in line 558) . It would be better to be more nuanced about the value that feedback from the coil and cantilever spring.

Wear and tear should be considered in the discussion.

Confirmation from the Institutional Review Board that ethics was not required is missing, as the paper does describe testing with a human subject.

The paper would be better positioned as a technical note.

Minor comments:

- General: please provide relevant units for each variable/parameter.

P1 - Suggesting to include the specific focus on sport in the title to better reflect the research conducted.

P1L16 - “Due to their prevalence and the fewer degrees of freedom in lower extremities, research and innovation have primarily focused on prosthetics for the legs and feet.” - I would question the correctness of this statement. Upper limbs have received a similar amount of interest as lower limbs in terms of (academic) research, as grasping provides interesting scientific questions. Based on the current literature I would expect that upper limb gets a disproportional amount attention compared to lower limbs when corrected for the number of users.

P4 – Figure 1 can be increased in size to help the reader better identify the different highlighted components of the test setup.

Reviewer 3 Report

The authors present an interesting and well written manuscript titled "Implementing Energy Return Elements in Upper Limb Prosthetics". 

The connection between modern body powered lower limb devices that leverage mechanical properties for improved energy return / performance is made for ways to improve upper limb devices. A novel 3D printing method to evaluate electromechanical properties of hybrid materials is interesting and reflects opportunities for force sensing or perhaps doping for customized response to loading. 

The manuscript is clear and well composed and will be of interest to the field as a framework for future customizable materials used in prosthetic designs. 

An additional paragraph discussing how the methods could impact both activity focused body powered (presently described) and general use myoelectric devices would be of interest. 

Author Response

We appreciate this suggestion and agree that it would be more interesting to discuss the impact of our approach with myoelectric-based prosthetics. We added a new paragraph as follows:

Section 6.1. Energy Return in Prosthetics and Beyond

[Line 644] Recent studies have yet to reach a conclusion as to which is superior between body-powered and myoelectric-based prosthetics. The former is well suited to applications that require a lot of energy, such as sports, and the latter is better optimized for fine adjustments, such as grasping an object [56]. In general cases, prosthetics are designed using one of the two methods, but a basketball prosthetic is a unique example requiring their combination. In our proposed prosthetics, the coil spring replicating the wrist motion is designed to provide more force, while the cantilever spring imitating the finger must be capable of finer adjustment of forces. More accurate basketball shots might be possible using a combination of energy return and other approaches, for instance if the force-sensing data from the cantilever can be incorporated into the myoelectric-based approach with an actuator.

Round 2

Reviewer 2 Report

The authors have addressed the questions adequately. However, it would be good to include a formal reference number with regards to the approval from the ethics committee. 

Author Response

Thank you very much for confirming our revision. Your feedback was very constructive and supportive in many ways. Through correspondence with Zach Pang, Managing Editor, we confirmed appropriate statements for the IRB, Informed Consent, and Conflict of Interest sections and updated them in the manuscript as follows:

Funding: This research was funded by Accenture, LLP.

Institutional Review Board Statement: The study (Accenture Basketball Prosthetic) was conducted in accordance with the legal and ethical requirements of Accenture, LLP.

Informed Consent Statement: Informed consent was obtained from all subjects involved in the study

Conflicts of Interest: Jung Wook Park, Taylor Tab, Eric Gallo, and Andreea Danielescu are currently and were employees of Accenture at the time research was done. Ben Greenspan was an employee of Accenture at the time work was completed but is now an employee of another company.